# Potential Use of COVID-19 Surgical Masks and Polyethylene Plastics in Developing Sustainable Concrete

Suvash Chandra Paul [1,*], Md. Ahosun Habib Santo [1], Sowmik Ahmed Nahid [1], Asifur Rahman Majumder [1], Md. Fahim Al Mamun [1], Md Abdul Basit [1] and Adewumi John Babafemi [2,*]

[1] Department of Civil Engineering, International University of Business Agriculture and Technology, Dhaka 1230, Bangladesh
[2] Department of Civil Engineering, Stellenbosch University, Stellenbosch 7602, South Africa
[*] Correspondence: suvashpl@iubat.edu (S.C.P.); ajbabafemi@sun.ac.za (A.J.B.);
Tel.: +01716601172 (S.C.P.); +27-21-808-4475 (A.J.B.)

**Abstract:** Managing disposable waste surgical face masks and plastic made from polyethylene (PE) resin is a real challenge. Thus, these are considered a great threat to the environment. Generally, surgical face masks are made of microplastic made of polypropylene materials. Both polypropylene and PE are not easily decomposable in the soil. Consequently, the presence of these waste materials can have detrimental effects on terrestrial and aquatic ecosystems, exacerbating the ongoing crisis faced by the animal kingdom and the broader biosphere. Hence, it is imperative to identify alternate and efficient methods for waste management. Given its significant economic importance, the construction industry holds a prominent position among many industries globally. Consequently, waste masks within the construction sector might assume a crucial role in mitigating plastic pollution. Concrete, one of the most widely used construction materials, is being adapted with various waste materials as the partial or complete substitutes for natural constituents, such as cement and aggregates. This study focused on using different percentages of used COVID-19 surgical masks in fiber form and PE as partial replacements of natural coarse aggregates in producing sustainable concrete. Mask fibers were used in concrete production at percentages of 0%, 0.5%, 1%, 1.5%, and 2% of the total volume of concrete. Similarly, PE aggregates replaced the coarse aggregates by volume at 0%, 5%, 10%, and 15% in concrete. The results showed that the strength of concrete reduced as the percentages of mask fiber and PE aggregates increased. However, the strength and crack-bridging capability of mask concrete are still acceptable for some structural and non-structural applications. The results obtained from this research could also help engineers to design sustainable concrete materials with mask fibers.

**Keywords:** facial masks; waste plastic aggregates; sustainable concrete; mechanical properties; electrical resistivity; water absorption; pore volume

## 1. Introduction

Plastics have become an essential part of our modern way of life, and the worldwide plastic generation has increased greatly during the past 50 years. This has contributed significantly to the generation of plastic-related waste. Plastic debris materials consist of excess, outdated, broken plastics of different household plastic materials (drink bottles, bottle caps, food wrappers, grocery bags, straws, etc.), tools, anti-static packaging materials, and devices made of plastic. In the past COVID-19 pandemic and the continuation of its impact in different countries these days, the production and use of single-use disposable facial masks, such as N95, cloth, fabric, surgical made of plastic, etc., increased suddenly all over the world in comparison to the past, to prevent the spread of COVID-19 viruses [1]. The masks under consideration comprise plastic materials, specifically polypropylene, polyethylene, and polyester. It is worth noting that these materials are currently recognized as significant contributors to the issue of plastic contamination [2]. The composition of

these entities consists of three distinct layers: an inside layer comprising pliable fibers, an intermediate layer consisting of a melt-blown filter, and an external layer composed of rigid fibers that possess water-resistant properties and are typically dyed [2]. During the COVID-19 pandemic period, it is reported that daily over 4 billion facial masks were used worldwide [3]. Prior to COVID-19, the global market for face masks was predicted to be around $0.73 billion, but it is now expected to exceed $22 billion [4]. Nevertheless, the utilization of masks and other forms of personal protective equipment (PPE), such as hand gloves and face shields, during the ongoing epidemic has resulted in significant environmental concerns on a global scale.

A significant quantity of discarded masks in various locations, such as roadsides, rivers, and canals, can be attributed to insufficient awareness and inadequate management practices. These masks pose a considerable risk to wildlife and marine life, as their non-biodegradable composition prevents natural decomposition [5]. Disposing waste face masks sensibly and avoiding environmental pollution is a new challenge for researchers. Mask waste collection and deposition in developing countries is a significant health risk, and the incineration method to dispose of these wastes is not highly recommended due to the toxic gases generated during the incineration of plastics [1]. Due to the presence of plastic constituents, the degradation or decomposition process of these disposable face masks is somewhat protracted. Therefore, the main problem is that these facial masks can destroy soil fertility and disrupt agricultural production, which may lead to extreme disaster in the agricultural sector.

In addition, there is a further problem with their adverse effects on the aquatic animals and the aquatic plants in rivers, drains, and canals, which disrupt the ecosystem. So, improperly managing these single-use waste face masks has become an environmental epidemic. Therefore, this is a complex problem, and simplifying it requires effective uses and environmentally friendly solutions that will work to prevent plastic pollution in the environment. The building industry is one of the economic sectors that uses a lot of natural resources and greatly influences the environment [6]. As a result, waste recycling has become one of the most important activities in the circular economy's long-term development [7,8].

To date, there has been a scarcity of studies conducted on the utilization of discarded masks in the manufacturing process of concrete [9,10]. A preliminary investigation was conducted to examine the effects of several percentages (0%, 2%, and 10% by weight of total mix composition) of shredded waste facial masks on manufacturing lightweight mortar intended for non-structural applications. It was concluded that the use of shredded waste facial masks is a viable approach for recycling in lightweight concrete as the compressive strength of the mortar increased from 1.14 MPa to 9.46 MPa and 19.45 MPa for 2% and 10% of shredded masks in the mixes [11].

In another study, different percentages of waste masks in concrete were used in two forms, such as in fiber form (at 1%, 1.5%, and 2%) and shredded (0.75%, 1%, and 1.5%), both by percentage weight of the cement [12]. At 28 days of testing, concrete samples made of fiber mask showed 0.36%, 5.73%, and 11.7%, respectively, higher compressive strength than the control samples without any mask. At the same age, for shredded mask samples, the strength increased by 3.58% and 18.28% for shredded mask content of 0.75% and 1%, but reduced by about 3.94% for 1.5% shredded mask content when compared with the control samples. Almost identical trends were noticed for the samples in the tensile strength test. The higher strength in mask concrete was attributed to better bonding between the matrix and mask. However, mixing was difficult at higher percentages of shredded mask and formed an inhomogeneous matrix; thus, strength was reduced [12]. The increment of strength with different percentages of masks in fiber form was also reported in the study [9]. At 28 days, the incorporation of masks at 0.5% and 1% of mix content demonstrated an increase in strength of approximately 8.3% and 17.9%, respectively, compared to the reference concrete. However, in the same study, the compressive strength of concrete was reported to have decreased by about 2% with the inclusion of 1.5% mask fiber. Higher



fiber content caused improper mixing due to interlocking of the fibers and resulted in inhomogeneity in the mixture. Therefore, the optimum fiber content in concrete is put at 1%.

In contrast to the previous studies, the compressive strength of concrete decreased as the percentage of shredded mask increased in the mixes [13,14]. Concrete samples with 0.5% shredded mask showed around 6% lower compressive strength than the control samples. Similarly, samples with a range of 1–3% shredded mask content showed around 20% lower strength; a maximum of 44% lower strength was found for samples with 5% shredded mask [13]. A mercury intrusion porosimetry test confirmed that as the percentages of mask content increased, the porosity in the samples also increased, which could lower the strength of the concrete [13].

The rapid chloride permeability test (RCPT) was conducted to assess the durability of concrete by varying the proportions of fiber mask content [9]. In concrete structures, a lower permeability to chloride ions is typically associated with enhanced resistance to steel corrosion [15,16]. Concrete samples with 0.5%, 1%, and 1.5% mask fiber showed lower chloride permeability than the control samples. A maximum decrease of 9.4% permeability was reported for the 1% fiber mask concrete [9]. Samples with 2% fiber content showed about 15% higher permeability than control samples. These findings indicate that including 1% fiber mask content increases air entrainment, leading to heightened permeability of the concrete. The ultrasonic pulse velocity (UPV) test was conducted on mortar samples, including shredded and strip masks. The results showed that as the percentage of mask material increased from 0.5% to 2%, the UPV value decreased by a minimum of 6% to 21% [14].

Researchers have also investigated the substitution of natural aggregates in concrete with varying proportions (measured by the weight of the concrete mix or the weight of the aggregates) of waste plastic aggregates composed of polyethylene terephthalate (PET) and polypropylene (PP) [17–19]. As the replacement level increased, concrete strength decreased gradually [17–19]. At 28 days of testing, the compressive strength of concrete was reduced by about 6%, 19%, and 36% for the replacement of natural coarse aggregates by waste PET aggregates of 10%, 20%, and 30%, respectively [17,18]. A similar trend of lower strength for PET concrete than the control concrete was reported in [18]. However, based on these investigations, the authors concluded that 10% PET in concrete could still meet strength requirements [17,18]. Another study compared the performance of the same percentages of PET and PP plastic aggregates [18]. It was reported that the concrete made of PP plastic aggregates showed more strength reduction than PET plastic concrete. The decline of strength could be attributed to their hydrophobic nature, thus providing less cohesion forces at the interface of aggregates and matrix, which forms poor bond strength. Also, the smooth surface of plastic aggregates may develop a weak interfacial transition zone (ITZ) between aggregates and matrix, leading to lower strength [20,21]. The slump of concrete was found to be increased as the percentage of plastic aggregates increased in the mix [17,18]. This is because the plastic aggregates are impermeable, and therefore, more water is available for the cement paste, which leads to higher workability. In the case of concrete density, the partial replacement of natural aggregates with plastic aggregates reduced the density of fresh and hardened concrete due to plastic's lower specific gravity and density than natural aggregates [19].

During the COVID-19 pandemic, $CO_2$ emissions were significantly reduced due to the lockdown in many countries and the closure of many industries, benefiting the environment [22]. Additionally, using different types of PPE during the pandemic effectively prevented the spread of viruses. However, the waste generated from the PPEs during the COVID-19 pandemic posed a great threat to the environment due to poor biomedical waste management [23]. In order to efficiently manage biomedical waste, numerous countries have implemented categorization systems based on the waste's sources and management methods [24,25]. However, it is important to note that this specific aspect falls outside the scope of this present study. Furthermore, the constrained availability of technology and a

deficiency in understanding effective waste management strategies have resulted in people adhering to conventional methods of disposal.

However, the ecological impact could potentially be mitigated to some extent through the recycling of this waste as an ingredient in a commonly utilized construction material such as concrete, following appropriate treatment procedures. While studies exist on the use of face masks and PE in concrete, there is a paucity of research on the workability and electrical resistivity of mask concrete. Other properties of mask and PE concrete have also been investigated. Hence, surgical face masks in the form of fibers are used to produce concrete in this research. Different percentages of fiber masks ranging from 0.5% to 2% by weight of concrete mix are used to produce concrete samples. Also, PE aggregate was used to replace the coarse aggregate by volume at 0%, 5%, 10%, and 15%. The concrete composite was tested for compressive strength, splitting tensile strength, water absorption, permeable pore volumes, and electrical resistivity. The microstructural analysis by scanning electron microscopy (SEM) was also performed on the concrete composite to investigate the effect of mask strips and plastic aggregates in the ITZ and the matrix. Although the strength of the concrete was reduced as the percentages of mask and PE aggregates increased, it might play a very important role in the development of aquatic, terrestrial, and plant life by preventing plastic contamination and developing a desirable eco-friendly environment. It is worth noting that new surgical face masks were used to prepare the mask fibers to avoid the possible spreading of COVID-19 via the use of used face masks. However, for the real application of used masks, proper technology must be adopted for the disinfection and sterilization process, which is out of the scope of this current study.

## 2. Materials and Methods

### 2.1. Materials and Mix Compositions

This study collected 3-Ply single-use surgical face masks approximately 175 mm long and 95 mm wide from the local market. The nasal wire metal frame and two ear straps from each mask were removed to ensure fiber material uniformity. After that, the masks were cut manually to have the desired geometric configurations of a median length of 42 mm and 3 mm width, respectively. It is worth mentioning that fiber over 40 mm is considered macro fiber, commonly used in concrete as plastic, synthetic, and steel fiber for improving concrete's tensile, flexural, shrinkage, cracking, and ductile behavior [26].

Plastic aggregate made of PE materials was also collected from the local recycling market, and the size ranged from 8 to 22 mm. Its water absorption capacity is 0.8%, and its specific gravity is 0.97. Figure 1 shows the aesthetic view of cut mask strips, SEM (Hi-TECH Instruments Sdn. Bhd., Selangor, Malaysia) images of mask morphology, and waste plastic aggregates used in this study. The typical physical properties of surgical masks examined by other researchers are reported in Table 1 [10]. CEM II/A-M 42.5 cement was used as the main binder of the concrete. This investigation employed coarse aggregates with a maximum size of 19 mm, which were processed from burnt clay brick, and sand with a maximum size of 4 mm as the fine aggregates. Figure 2 presents the sieve analysis of both coarse aggregates (CA) and fine aggregates (i.e., sand) along with plastic aggregates (PA) used for concrete mixes. The mask strip fibers were used at 0%, 0.5%, 1%, 1.5%, and 2% by volume of the total concrete weight; the mixes are identified as MC1, MC2, MC3, MC4, and MC5 in Table 2. Also, the plastic aggregate, PE, was used as a replacement for the coarse aggregate at 0%, 5%, 10%, and 15% by volume, and the resulting mixes are defined in Table 2 as PC1, PC2, PC3, and PC4. A total of 9 mixes (5 from mask and 4 from plastic) were designed for this research work. The mix compositions of the different concrete mixes are also presented in Table 2. It must be noted that no treatment methods were followed for modifying the surface of the mask and plastic aggregate used. For the proper distribution, mask strips were mixed with the dry aggregates and the binder before adding the water. The total mixing time for both mixes was around 5−6 min. All aggregates were used in air-dry conditions to replicate the mass concrete mixing for real construction work.

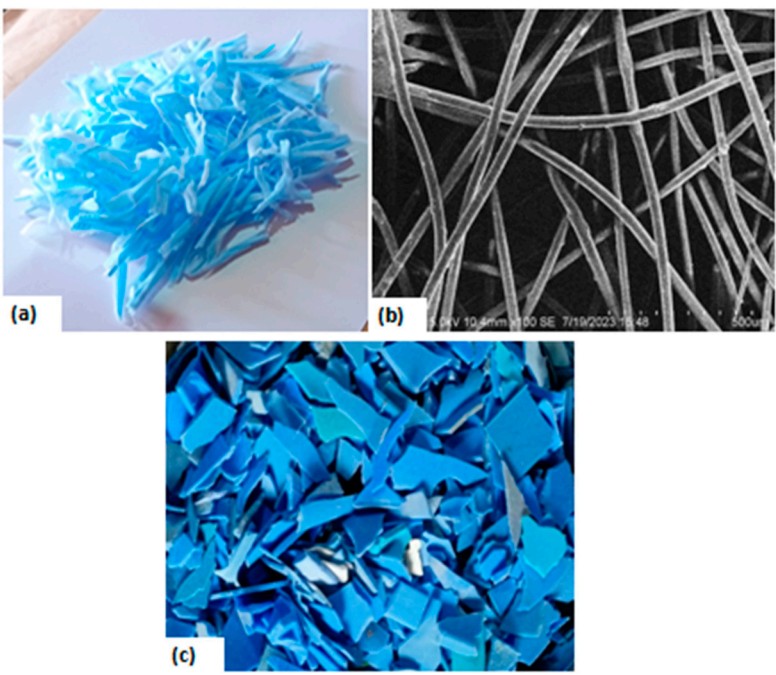

**Figure 1.** (**a**) Mask strip fiber, (**b**) SEM images of mask strip morphology, and (**c**) waste plastic aggregates used in this study.

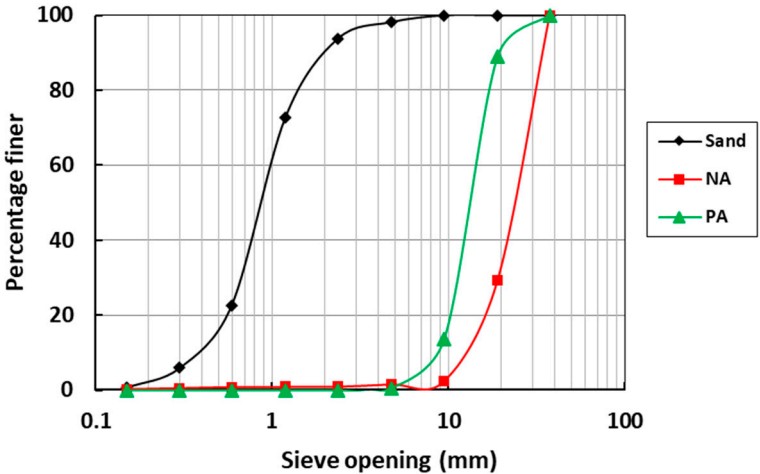

**Figure 2.** Sieve analysis of sand, coarse aggregate (CA), and plastic aggregate (PA).

**Table 1.** Typical physical properties of the single-use surgical mask [10].

| | |
|---|---|
| Specific gravity | 0.91 |
| Melting point | 160 °C |
| Water absorption | 8.9% at 24 h |
| Tensile strength at breaking point | 3.97 MPa |
| Elongation at break | 119% |

**Table 2.** Mix compositions of different concrete mixes made of mask strips and plastic aggregate in kg/m$^3$.

| Mix ID | Cement | CA | Sand | Water | Mask Strip | PA |
|--------|--------|-----|------|-------|------------|-----|
| MC1 | 463.35 | 1094.5 | 739.84 | 208.51 | 0 | - |
| MC2 | 460.84 | 1088.56 | 735.82 | 207.38 | 5.25 | - |
| MC3 | 458.35 | 1082.68 | 731.85 | 206.26 | 10.46 | - |
| MC4 | 455.89 | 1076.87 | 727.92 | 205.15 | 15.6 | - |
| MC5 | 453.45 | 1071.12 | 724.03 | 204.05 | 20.69 | - |
| PC1 | 446.8 | 969.23 | 645.38 | 201.06 | - | 0 |
| PC2 | 418.32 | 907.45 | 604.24 | 188.24 | - | 61.73 |
| PC3 | 392.98 | 852.47 | 567.63 | 176.84 | - | 116.66 |
| PC4 | 370.89 | 804.57 | 535.74 | 166.9 | - | 164.52 |

Note: CA is coarse aggregate, PA is plastic aggregate, MC and PC mean concrete made of mask strips and plastic aggregates.

*2.2. Samples Preparation and Testing Methods*

After mixing, a slump test was performed to check the workability of the fresh concrete. For compressive and splitting tensile strength of the concrete, 100 mm × 100 mm × 100 mm cube samples were prepared according to the BS standards [27,28]. All mixing was performed at laboratory temperature, and the fresh concrete was poured into the molds layer-by-layer and tampered (i.e., for compaction and removing the air bubbles) according to the standards. The fresh concrete samples were left in the molds for at least 24 h before demolding. After that, the samples were water tank-cured for 7, 14, and 28 days. The compressive and splitting tensile strength tests were performed in a universal testing machine (UTM) (Zhejiang Tugong Instrument Co., Ltd., Shangyu, China) where the loading rate was applied at about 4 kN/s. A minimum of four samples for each day of testing, thus, a total of 196 samples (100 for compression and 96 for splitting tensile strength), were prepared. In addition, 100 mm diameter and 50 mm thickness concrete disk samples were also prepared for water absorption and permeable-pore volume testing of all mixes according to the ASTM C642 [29]. In this case, three samples were considered for each mix, and the test was performed only after 28 days of curing. Equations (1) and (2) were used to calculate the absorption and the pore volume in the samples.

$$\text{Absorption after immersion (\%)} = [(m_2 - m_1)/m_1] \times 100 \tag{1}$$

$$\text{Permeable pore volume (\%)} = [(m_3 - m_1)/(m_3 - m_4)] \times 100 \tag{2}$$

where $m_1$, $m_2$, $m_3$, and $m_4$ (g) represent the sample mass in oven-dried, surface dried in the air after immersion in the water, surface dried in the air after immersion and boiling in the water, and the apparent mass of the sample in water after immersion and boiling, respectively. The sample preparation and testing procedures are broadly discussed in ASTM C642 [29].

The electrical resistivity (ER) of the concrete samples was also measured on the same specimens tested for absorption and pore volume. It is a non-destructive test usually used to assess the concrete performance against chloride ion movement linked to corrosion of the steel bars. All samples were subjected to saturated surface dry (SSD) conditions, after which an LCR meter (Extech Instruments Corporation, Nashua, NH, USA) was used to measure the resistance of the samples by applying frequencies of 100 Hz, 1 kHz, and 10 kHz to the two surfaces of the disc samples. The ER of the samples was then calculated using Equation (3) below.

$$\text{ER} = \frac{A}{L} R \tag{3}$$

where, R, A, and L are the resistance, area, and length of the sample.

Furthermore, SEM analysis assessed the ITZ between cement and other substances used to produce concrete in this study. The SEM images were collected at the 10–15 kV mode of the microscope, and the area of the images was between 1000 μm$^2$ to 3000 μm$^2$.

## 3. Results and Discussion

### 3.1. Workability of Mask and Plastic Concrete

The slump test is used to determine the consistency, also known as the workability, of concrete. The workability of concrete plays an important role in ensuring concrete strength and durability. Collapse of slump indicates the concrete mix is too wet, and the mix is regarded as harsh and lean. Very dry mixes having slump 0–25 mm are typically used in constructing pavements or roads, low workability mixes having slump 25–50 mm are typically used for foundations with light reinforcement, and medium workability mixes with slump 50–10 mm are typically used for normal reinforced concrete placed with vibration [30].

Figure 3 shows the slump of concrete made of mask strips and plastic aggregate. In almost all cases (except for MC2), the workability reduces as the mask and plastic content percentages increase. For mask concrete, when compared to the reference concrete (i.e., MC1), the reduction in workability is 0%, 40%, 60%, and 80% for the inclusion of mask strips of 0.5%, 1%, 1.5%, and 2%, respectively. In the case of plastic aggregate, these reductions are 5%, 14%, and 21% for the replacement of 5%, 10%, and 15% coarse aggregates by the plastic aggregates. As far as the authors know, no research has investigated the workability of concrete with mask strips. Consequently, it is not feasible to make direct comparisons between the findings of this study and the existing literature. However, the reduced slump of mask concrete could be attributed to increased water absorption with the increased percentage of the mask strips in concrete. The air-dried mask strips exhibited a larger capacity for water absorption, resulting in a greater amount of water being retained by these strips and consequently lowering water availability for the other constituents of the concrete mixture. This reduction in water availability ultimately led to a decrease in the workability of the concrete. Similar trends are also found for plastic aggregate concrete. The non-uniform shapes and thinner size of plastic aggregate prevented the fluidity of the concrete mixes. Nevertheless, despite the reduced workability of mask concrete, plastic aggregate provides better workability than mask concrete since lower water absorption of plastic can increase concrete flow [31].

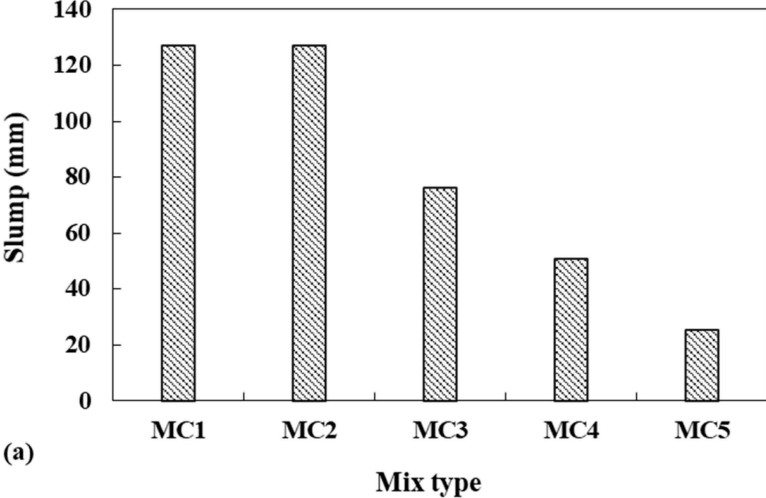

**Figure 3.** *Cont.*

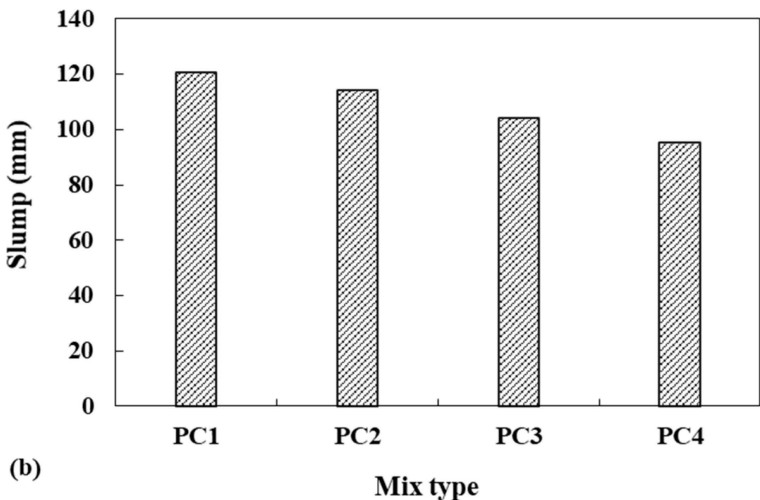

(b)

**Figure 3.** Slump values of (**a**) mask concrete (MC) and (**b**) plastic concrete (PC).

The yield stress of concrete samples was calculated from the empirical formula proposed in the existing literature [32,33], as shown in Equations (4) and (5). As seen in Figure 4, the maximum yield stress is obtained for mask concrete (see Figure 4a) and decreases as the slump value reduces. The differences in the yield stress for mask concrete are also significant. In contrast, it is insignificant for plastic concrete, as shown in Figure 4b. The differences in the results are also noticeable for the different formulas used. This is because both are empirical formulas considering different constant parameters in predicting the yield stress from the slump values of concrete.

$$\text{s} = 300 - 347 \, \frac{(\tau_o - 212)}{\rho} \tag{4}$$

$$\text{s} = 25.5 - 17.6 \, \frac{\tau_o}{\rho} \tag{5}$$

where s is the slump, $\tau_0$ is the yield stress, and *p* is the concrete density.

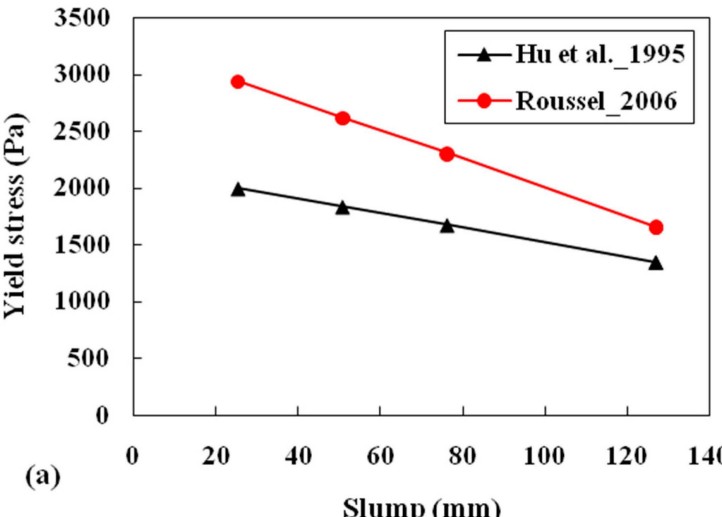

(a)

**Figure 4.** *Cont.*

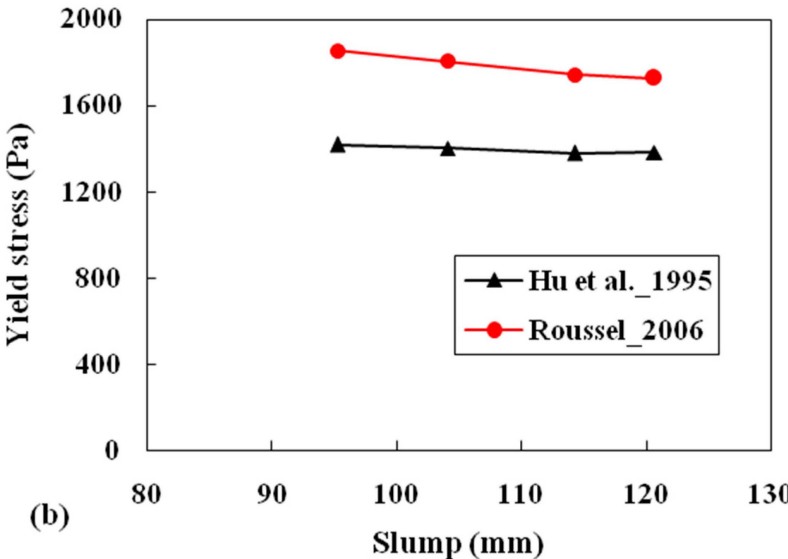

**Figure 4.** Estimated yield stress of fresh (**a**) mask concrete and (**b**) plastic concrete [32,33].

### 3.2. Dry Density of Concrete Samples

The dry densities of the hardened concrete samples are shown in Figure 5. Generally, the mechanical properties of concrete are linked to its density. Higher density can contribute to higher concrete strength. Although other factors, such as the properties of aggregates, methods of curing, and compaction of fresh concrete, also play a role in the mechanical properties of concrete. According to the data presented in Figure 5a, incorporating 2% mask strips reduced the dry density by about 19% compared to the control samples of concrete. In the context of plastic aggregate concrete, substituting 15% of natural coarse aggregates with plastic aggregates resulted in a reduction of approximately 15% in density compared to the reference concrete samples. Overall, the lower density of the mask strip and plastic aggregate concrete can reduce the weight of concrete structures, thus lowering the construction cost.

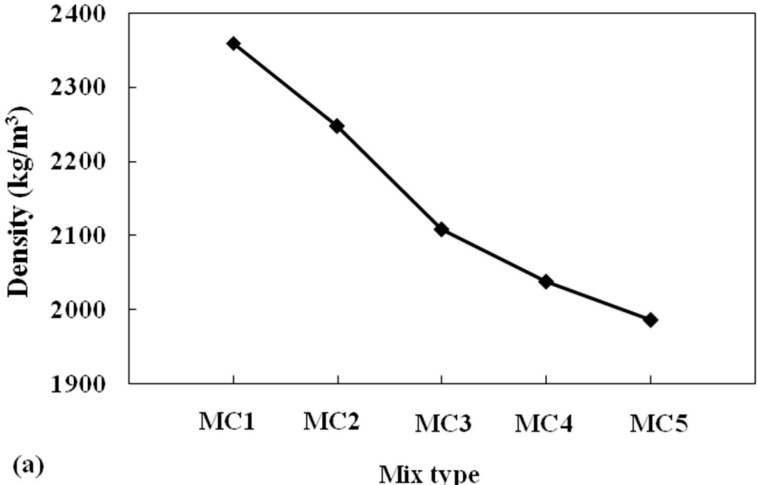

**Figure 5.** *Cont.*

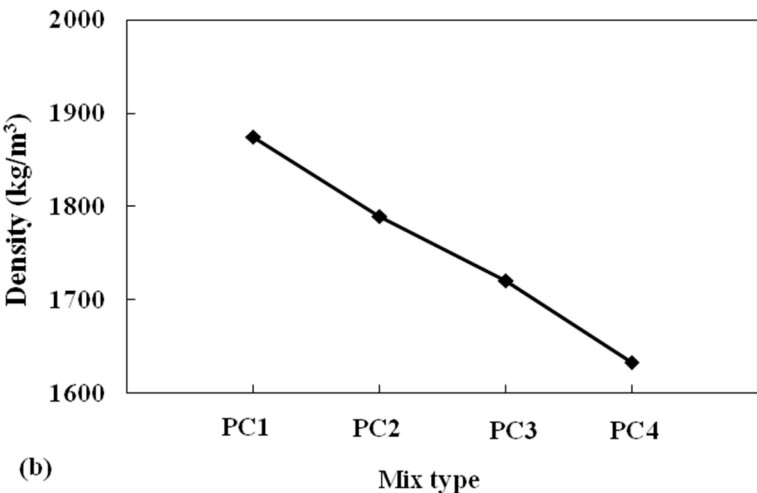

**Figure 5.** Dry density of (**a**) mask strip and (**b**) plastic aggregates concrete.

### 3.3. Mechanical Strength of Mask and Plastic Concrete

The development of compressive strength in the mask and plastic concrete samples is presented in Figure 5. The strength gradually decreases with an increase of mask strip and plastic aggregate content in the mixes. For the mask concrete, as depicted in Figure 6a, after 28 days of curing, the reduction of the compressive strength is about 20%, 31%, 57%, and 58% for MC2, MC3, MC4, and MC5, respectively, in comparison to the MC1 mix. In the case of plastic concrete, these reductions are 13%, 32%, and 52% for PC2, PC3, and PC4 when compared to the PC1 mix.

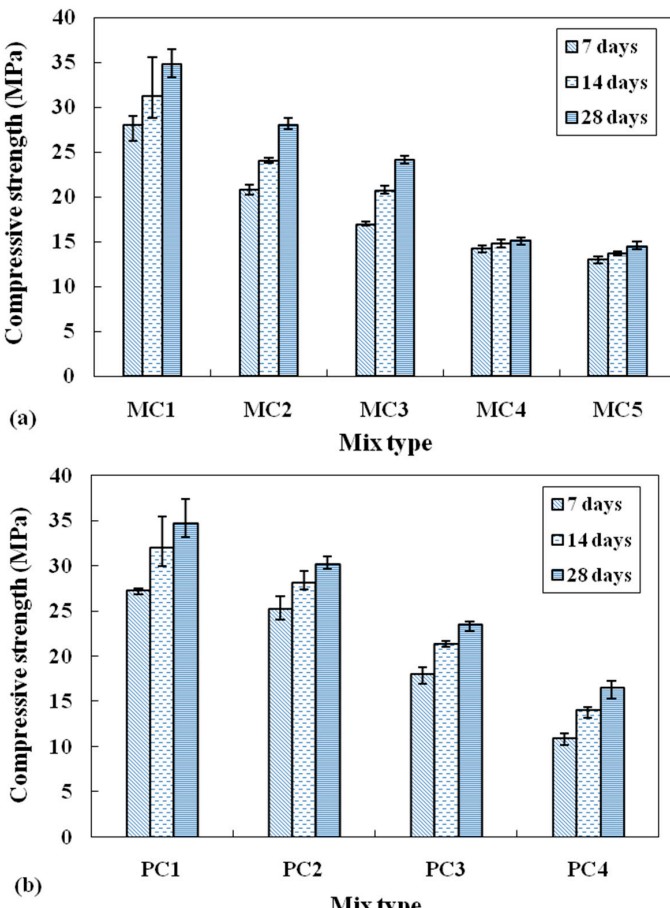

**Figure 6.** Compressive strength developments in (**a**) mask concrete (MC) and (**b**) plastic concrete (PC).

Figure 7 shows the splitting tensile strength results of MC and PA concrete samples tested at different curing ages. Similar to the compressive strength, the tensile strength of the concrete also reduces as the mask strip and PA percentages increase in the mixes. At 28 days, the strength is reduced by 11%, 18%, 24%, and 31% for MC2, MC3, MC4, and MC5 compared to the MC1 concrete. In the case of PA concrete, these reductions are 10%, 32%, and 37% for PC2, PC3, and PC4 compared to the PC1 concrete.

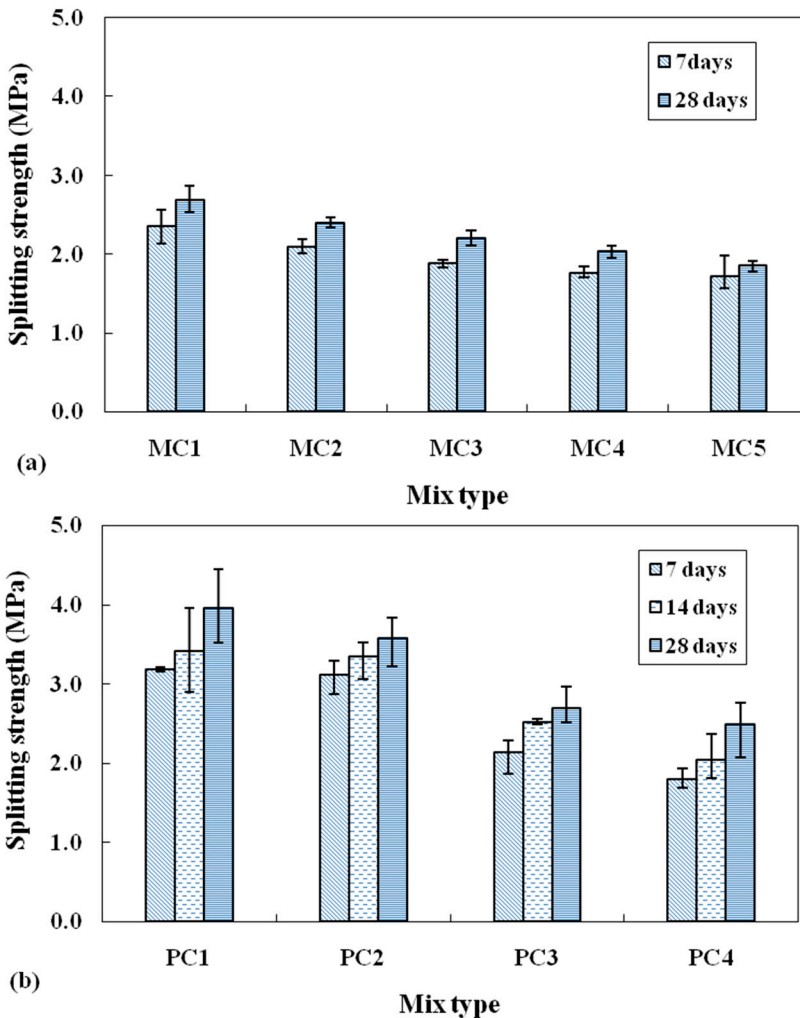

**Figure 7.** Splitting tensile strength in (**a**) mask concrete (MC) and (**b**) plastic concrete (PC) across different days.

The experimental results obtained from this research are also compared with the existing literature [34–40] for similar types of materials and their replacement level, as illustrated in Figure 8. Figure 8a shows the comparison of the normalized compressive strength results (i.e., concrete strength with fiber percentages divided by the reference concrete strength without any fiber) of this study's mask strip concrete with existing results in the literature including concrete mixed with various fibers such as metallic plastic (MP), plastic fiber (PF), wire plastic fiber (WPF), PET, and PP. It can be seen that the reduction of the strength of mask concrete follows the same trend as of other types of fibers, such as MP, PF, and PET investigated by previous researchers. However, the reduction rate in strength is found to be higher in mask concrete than others, as summarized in Figure 8a. However, some steps could be followed in future research work to minimize the strength reduction while using the mask strips, such as treating the surface of the mask strips to reduce the voids and water absorption. In this regard, some epoxy can be used to fill up the voids in the mask strips, as shown in Figure 1b. Another option could be optimizing

the mix compositions so that the strength of concrete does not reduce significantly while using certain percentages of mask strips.

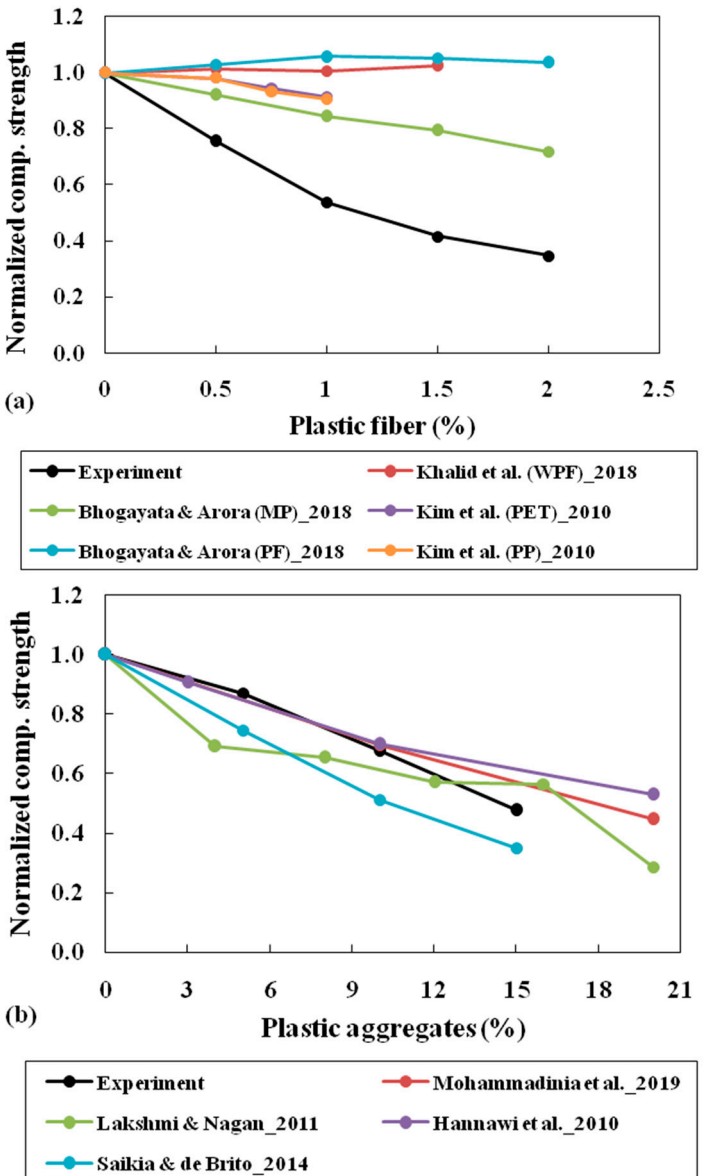

**Figure 8.** Normalized compressive strength of (**a**) plastic fiber and (**b**) plastic aggregates concrete [34–40].

Figure 8b summarizes the normalized compressive strength of the experimental results of plastic aggregate concrete from this research along with the other research available in the literature. Although the strength is reduced as the percentages of plastic content increase, the results follow a similar trend of strength reduction, which other researchers have reported.

### 3.4. Water Absorption and Pore Volume in Mask and Plastic Concrete

The inclusion of masks and plastic in concrete contributed to increased water absorption and pore volume in the mixes, as reported in Figure 9. These properties depend on the concrete mix composition, such as water-to-binder ratio and cement paste volume. In the case of mask concrete, including 0.5–2% mask strips increased the absorption and pore values about 22–70% and 12–70% when compared to the reference mix (MC1). For plastic concrete, these values were 19–36% and 32–40% higher than PC1 for the inclusion of 5–15% plastic aggregate in concrete. The hydrophobic nature of plastic materials may lead to weak

zones around the contact surface with the cement paste and thus allow water or liquid to enter.

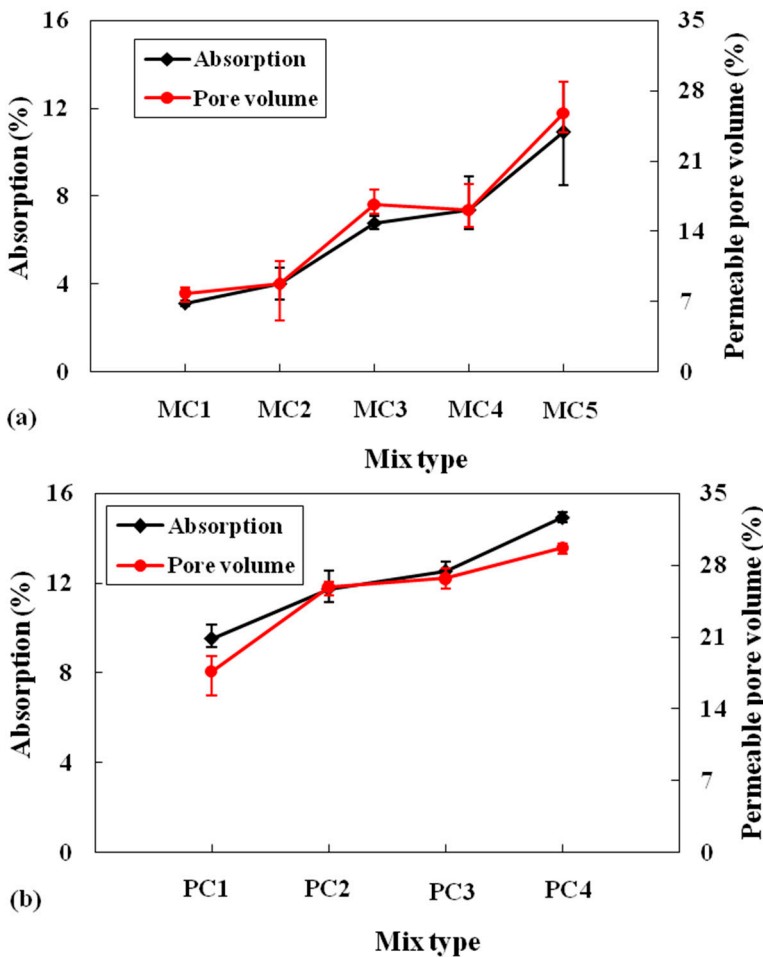

**Figure 9.** Percentages of water absorption and pore volume in (**a**) mask concrete (MC) and (**b**) plastic concrete (PC) samples after 28 days of curing.

### 3.5. Electrical Resistivity of the Concrete Samples

The ER values of both concrete types (MC and PC) are shown in Figure 10. Generally, at higher frequency levels, ER values in the samples are reduced, which is also observed in this study. The ER values gradually decrease as the mask strip content increases in the mixes, as presented in Figure 10a. At 1 kHz frequency level, the ER values of the samples are reduced by 48%, 52%, 52%, and 99% with the inclusion of 0.5%, 1%, 1.5%, and 2% mask strips in the concrete mixes. Higher pore volumes in the concrete samples due to the inclusion of mask strips may lead to lower resistivity of the concrete samples. As far as the authors know, to date, there is no research that has reported on the resistivity of concrete with mask strips. For plastic aggregate concrete, an opposite behavior of ER values is observed. In this case, the ER values at lower frequencies (100 Hz and 1 kHz) increase as the plastic aggregate content increases in the mix. Although the pore volume in this concrete also increases as the plastic content increases, there seems to be no relationship with the resistivity of the concrete. A study on the chloride ingress in plastic aggregate concrete also reported similar behavior of better performance than the control mix concrete [41]. The total charge passed through concrete samples is reduced as the plastic aggregate content increases. This behavior of plastic aggregate concrete could be attributed to the impervious nature of plastic aggregate, which can create a physical obstacle for the charge passing through the samples [42]. At 15% replacement level with plastic aggregates, the ER increased to a

maximum of 52% and 33% for 100 Hz and 1 kHz applied frequencies when compared with the control samples of concrete mix.

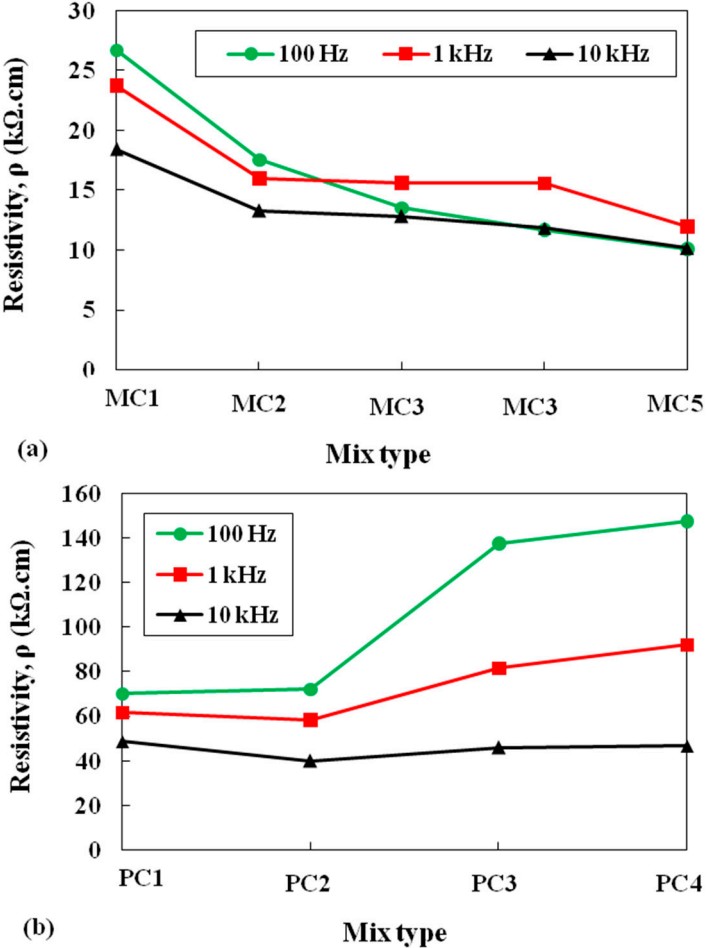

**Figure 10.** Resistivity of concrete samples with (**a**) mask strips and (**b**) plastic aggregates.

The relationships between the absorption, pore volume, and resistivity of the concrete samples were established, as illustrated in Figure 11. For mask concrete, the resistivity increases as the absorption and pore volume reduces, as shown in Figure 11a. However, this behavior is opposite for the plastic aggregate concrete in Figure 11b. In this case, the resistivity increases as the absorption and pore volume increases. The impervious nature of plastic aggregate may create obstacles for the electrical current passing through the concrete samples regardless of the higher pore volume. Further studies would be required to validate these results as there is currently limited research on the resistivity of mask concrete. It is also worth mentioning that the electrical resistivity measuring technique adopted by the researchers and the standards are mainly for conventional concrete without any substances such as plastic, glass, and iron slag aggregates. Therefore, the suggested electrical resistivity values for conventional concrete may need to be revised for concrete samples made from ingredients other than natural aggregates.

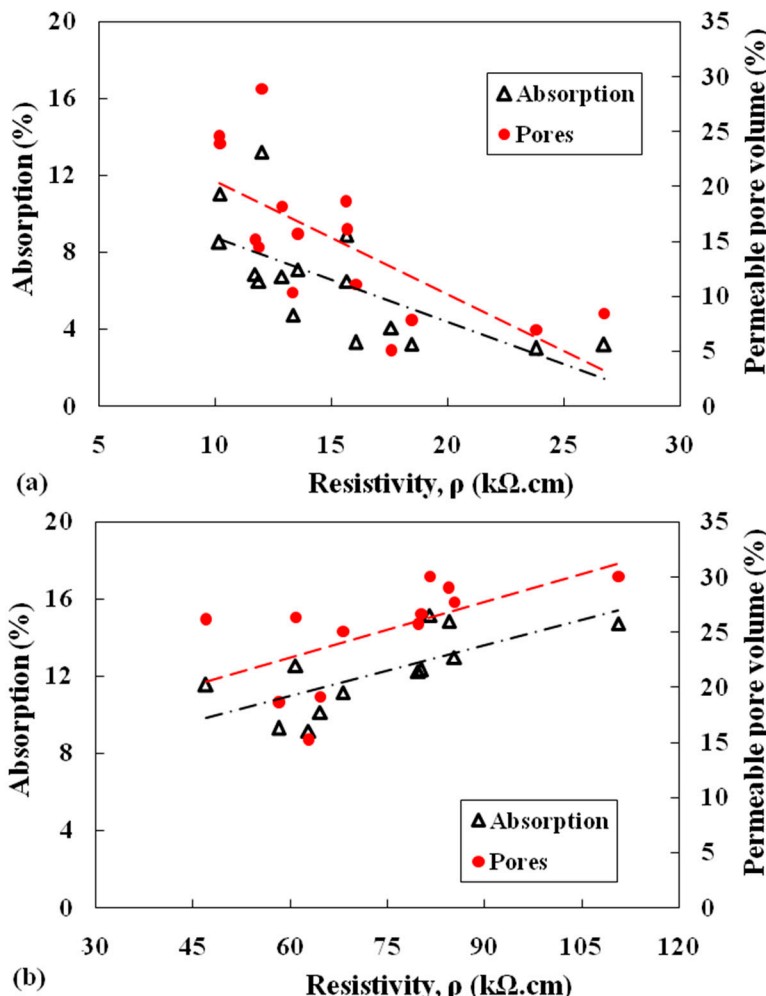

**Figure 11.** Relation of resistivity to the absorption and pore volume of the (**a**) mask strips and (**b**) plastic aggregate concrete samples.

*3.6. Microstructural Analysis of Concrete*

The SEM analysis for the microstructure of the different hardened concrete mixes (PC1, PC3, and MC3) is presented in Figure 12. An improved ITZ is found for the reference concrete mix of PC1, as shown in Figure 12a. The inclusion of 10% PA significantly affects the ITZ of PC3 concrete samples, as shown in Figure 12b. In this case, a visible crack is observed between the cement paste and PA. This could be attributed to the hydrophobic nature of plastic, which may weaken the bond strength between the cement paste and the surface of PA. Introducing mask strips also leads to weak microstructure in concrete, as shown in Figure 12c. Since mask strips contain multiple fibers together, as reported in Figure 1b, it can form pores in the matrix and lead to lower strength and higher water absorption, as reported in Figures 6 and 9.

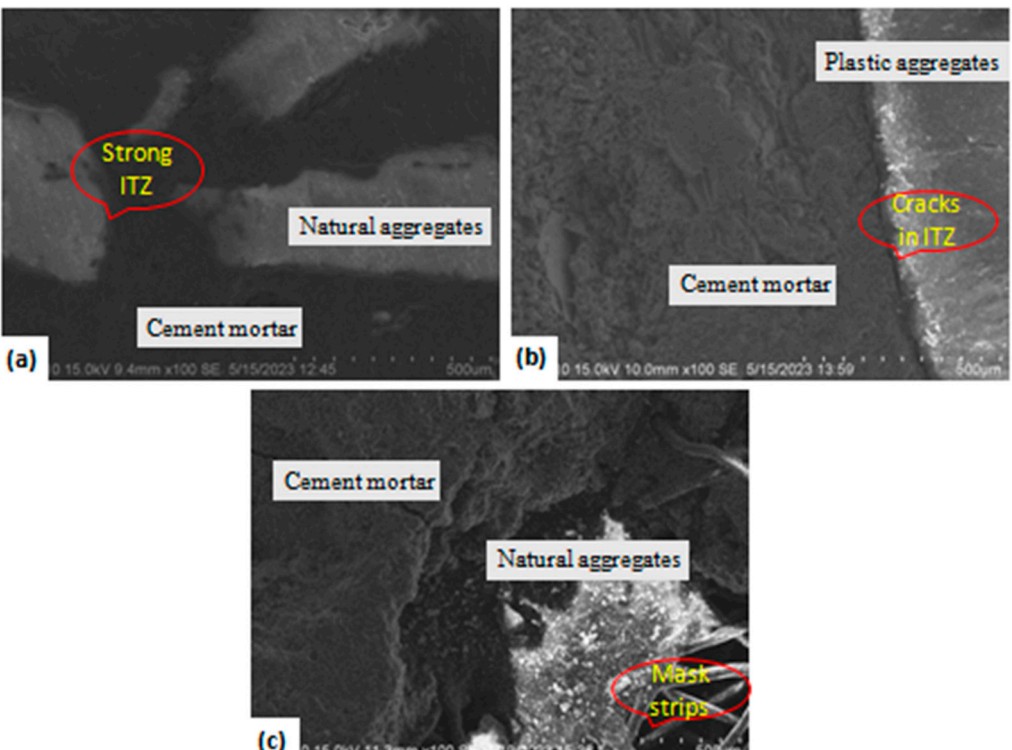

**Figure 12.** SEM analysis for the microstructure of (**a**) PC1, (**b**) PC3 concrete samples, and (**c**) MC3 concrete samples.

## 4. Conclusions and Recommendations

The utilization of waste materials constitutes a crucial aspect of the circular economy, garnering significant attention from sustainability policymakers across the globe. An enormous amount of masks and plastic are converted to waste daily, and using them in concrete could save the environment to some extent. A reduction in concrete density can be achieved by using specific proportions of masks or plastic materials. The decreased density of the concrete constructions may result in a corresponding reduction in self-weight, potentially leading to cost savings. To this aim, this study was conducted, and the following key findings can be summarized from the experimental research work conducted:

Higher surface roughness of both mask strips and plastic aggregates could lower the workability of concrete. A maximum reduction of 80% and 21% in the workability of concrete are recorded for a 2% mask strip and 15% plastic aggregate, respectively.

The concrete strength reduced as mask strips and plastic aggregate content increased in the mixes. However, compared with the compressive strength, the rate of tensile strength reduction is lower for both mask and plastic concrete. Moreover, 5% plastic aggregate seems acceptable as the mechanical properties of concrete with this amount of plastic did not alter much.

Adding waste masks and plastic increases the permeable pore volume and absorption percentages in the concrete samples. Additionally, 0.5% mask strips in concrete did not affect concrete samples' absorption and permeable pores.

A good correlation is found in the electrical resistivity of mask strip concrete, i.e., absorption and pore volume increased, and the resistivity of the samples decreased. However, the impervious nature of plastic aggregates showed the opposite behavior compared to the mask strip concrete.

The hydrophobic nature of plastic aggregate forms weak bonds, thus initiating cracks in the ITZ, leading to lower strength of the concrete samples.

Though both masks strips and plastic aggregate negatively affect the properties of concrete, they can still produce a level of concrete strength which may be sufficient for

some applications such as paver blocks, temporary structures, and non-structural elements. Furthermore, research would be required to gain in-depth knowledge of the different micromechanical and durability performances of concrete with these waste substances.

**Author Contributions:** Conceptualization, S.C.P.; methodology, S.C.P., M.A.H.S., S.A.N., A.R.M. and M.F.A.M.; validation, S.C.P., M.A.B. and A.J.B.; formal analysis, S.C.P.; investigation, S.C.P., M.A.H.S., S.A.N., A.R.M. and M.F.A.M.; data curation S.C.P., M.A.B. and A.J.B.; writing—original draft preparation, S.C.P.; writing—review and editing, M.A.B. and A.J.B.; supervision, S.C.P. All authors have read and agreed to the published version of the manuscript.

**Funding:** This research received no external funding.

**Data Availability Statement:** Data can be obtained from the corresponding authors upon request.

**Acknowledgments:** The authors would like to thank the Miyan Research Institute of the International University of Business Agriculture and Technology for supporting this research. The authors would also like to thank Md. Arif Hossain, Riaz Mahmud and Amirul Islam Turja for helping out with the experiments.

**Conflicts of Interest:** The authors declare no conflict of interest.

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
