# Peer review of "Potential Use of COVID-19 Surgical Masks and Polyethylene Plastics in Developing Sustainable Concrete"

_jcs, doi:10.3390/jcs7090402_

Round 1

Reviewer 1 Report

1.None of the typical physical properties of waste plastic aggregates used in the study.

2.Plastic aggregate was used as a replacement for coarse aggregate at 0%, 5%, 10%, and 15% by volume, however, it doesn't provide its specific gravity.

3. What is the difference between MC1 and PC1 concrete mix compositions in Table 2?

4. How can the use of COVID-19 surgical masks and PE plastic in concrete be sustainable with no instructions regarding future recycling.

5. Figure 4 is based on the relationship between concrete slump and yield stress proposed in the existing literature [30, 31]. It should be calculated and understood only by presenting its empirical formula.

Author Response

Title: Potential use of COVID-19 surgical masks and PE plastics in developing sustainable concrete

Dear Reviewer:

Thanks for your comments on our manuscript titled “Potential use of COVID-19 surgical masks and PE plastics in developing sustainable concrete”. The comments are all valuable and helpful for improving the manuscript and even instructive for our future research. Accordingly, we have studied these insightful comments carefully and revised our manuscript according to your suggestions to meet the journal standard. The main corrections and the response to the comments are as follows.

Reviewer 1:

1. None of the typical physical properties of waste plastic aggregates used in the study.

Authors Reply: Thank you for your comments. The water absorption, specific gravity and fineness modulus of plastic aggregates are added in the revised manuscript.

2. Plastic aggregate was used as a replacement for coarse aggregate at 0%, 5%, 10%, and 15% by volume, however, it doesn't provide its specific gravity.

Authors Reply: Specific gravity is now added.

3. What is the difference between MC1 and PC1 concrete mix compositions in Table 2?

Authors Reply: MC and PC mean concrete made of mask strips and plastic aggregates. It is now also added to the footnote of Table 2.

4. How can the use of COVID-19 surgical masks and PE plastic in concrete be sustainable with no instructions regarding future recycling?

Authors Reply: Thank you for your comment. The utilization of waste materials constitutes a crucial aspect of the circular economy, garnering significant attention from sustainability policymakers across the globe. An enormous amount of masks and plastic are converted to waste daily and using them in concrete could save the environment to some extent. A lower density of concrete could be obtained by using certain percentages of masks or plastic in it. This lower density may cause lower self-weight of the concrete structures and can reduce the cost. This is now discussed in the revised manuscript.

5. Figure 4 is based on the relationship between concrete slump and yield stress proposed in the existing literature [30, 31]. It should be calculated and understood only by presenting its empirical formula

Authors Reply: Thank you for your comments. We have now added the empirical formula in the revised manuscript. Please see Equations 4 and 5.

Reviewer 2 Report

Review of jcs-2598902

This is an interesting manuscript about one of the utilization of waste face masks that surge during COVID-19, as a mixture for concrete. The concrete samples in this manuscript were properly characterized. However, several issues of this manuscript must be kindly addressed, as follows:

  1. Section 1: Please kindly add more references about waste management during COVID-19 in several countries, such as: 
  • COVID-19 waste management in China: Sustainability 14(8) 2022 4746 https://doi.org/10.3390/su14084746
  • COVID-19 waste management in Indonesia: Sustainability 14(5) (2022) 2556 https://doi.org/10.3390/su14052556
  • COVID-19 waste management in India: Environmental Science and Pollution Research 28 (2021) 52702-52723 https://doi.org/10.1007/s11356-021-15028-5
  • COVID-19 waste management in Malaysia: Waste Management and Research 39 (2021) 18-26 https://doi.org/10.1177/0734242X20959701

  1. Line 53: Please change “Covid-19” to be “COVID-19”
  2. Figure 2: Please kindly convert this figure to be the bell curve of pore size distribution. Please explain what do you mean with “Percentage finer” in the y-axis of Figure 2.

  1. Figure 8: The result of this work in Figure 8 is the significant drop of compressive strength to be only 40% of the initial compressive strength, just by the addition of 2% plastic fiber. On the other hand, the addition of plastic aggregates also reduces the compressive strength to around 50%, but at much higher addition of plastic aggregates of 15%, which is still can be appreciated in waste utilization. However, the concretes are highly susceptible to be weakened simply by the addition of only 2% plastic fiber. Please kindly state about what strategies that must be taken in order to obtain better and more encouraging results in the future.

  1. Figure 10a: Please write the unit of resistivity “kΩ.cm” with lowercase k (not uppercase K). Please write the unit of frequency as kHz, with lowercase k, uppercase H, and lowercase z.
  2. Figure 10b: Please write the unit of resistivity “kΩ.cm” with lowercase k (not uppercase K). Please write the unit of frequency as kHz, with lowercase k, uppercase H, and lowercase z.
  3. Figure 11a: Please write the unit of resistivity “kΩ.cm” with lowercase k (not uppercase K).  
  4. Figure 11b: Please write the unit of resistivity “kΩ.cm” with lowercase k (not uppercase K).  
  5. Figure 10 and Figure 11: Please write the relation and connection of the results in Figure 10 and 11, such as the relation of resistivity to the absorption and pore volume.
  6. Figure 10 and Figure 11: Please also elaborate on why the trend of Figure 10a (decreasing) is “contradicting” to that of Figure 10b (increasing), similar to that of Figure 11a (decreasing)  that is “contradicting” to that of Figure 10b (increasing).
  7. Figure 11a: Please make the y-axis to be ranged from 0 to 20%, identical to that of Figure 11b, in order to obtain quick comparison between both of them.

Author Response

Title: Potential use of COVID-19 surgical masks and PE plastics in developing sustainable concrete

Dear Reviewer:

Thanks for your letter and comments for our manuscript titled “Potential use of COVID-19 surgical masks and PE plastics in developing sustainable concrete”. The comments are all valuable and helpful for improving the manuscript and even instructive for our future research. Accordingly, we have studied these insightful comments carefully and revised our manuscript according to your suggestions to meet the journal standard. The main corrections and the response to the comments are as follows.

Reviewer 2:

This is an interesting manuscript about one of the utilization of waste face masks that surge during COVID-19, as a mixture for concrete. The concrete samples in this manuscript were properly characterized. However, several issues of this manuscript must be kindly addressed, as follows:

1. Section 1: Please kindly add more references about waste management during COVID-19 in several countries, such as: 

  • COVID-19 waste management in China: Sustainability 14(8) 2022 4746 https://doi.org/10.3390/su14084746
  • COVID-19 waste management in Indonesia: Sustainability14(5) (2022) 2556 https://doi.org/10.3390/su14052556
  • COVID-19 waste management in India: Environmental Science and Pollution Research28 (2021) 52702-52723 https://doi.org/10.1007/s11356-021-15028-5
  • COVID-19 waste management in Malaysia: Waste Management and Research39 (2021) 18-26 https://doi.org/10.1177/0734242X20959701

2. Line 53: Please change “Covid-19” to be “COVID-19”

 Authors Reply: Thank you for your comment. Section 1 is improved by adding relevant literature as suggested by the reviewers.

3. Figure 2: Please kindly convert this figure to be the bell curve of pore size distribution. Please explain what do you mean with “Percentage finer” in the y-axis of Figure 2.

 Authors Reply: Figure 2 shows the sieve analysis of the different aggregates used in the study. It is not related to the pore size distribution. Percentage finer is achieved from the standard sieve analysis of aggregate for calculating the fineness modulus.

4. Figure 8: The result of this work in Figure 8 is the significant drop of compressive strength to be only 40% of the initial compressive strength, just by the addition of 2% plastic fiber. On the other hand, the addition of plastic aggregates also reduces the compressive strength to around 50%, but at much higher addition of plastic aggregates of 15%, which is still can be appreciated in waste utilization. However, the concretes are highly susceptible to be weakened simply by the addition of only 2% plastic fiber. Please kindly state about what strategies that must be taken in order to obtain better and more encouraging results in the future.

Authors Reply: Thank you for pointing out it. We have now discussed the strategies that could be taken in future to obtain better and more encouraging results in future in the revised manuscript.

Some steps could be followed in future research work to minimize the strength reduction while using the mask strips such as treating the surface of the mask strips using the epoxy. This fills the voids thus water absorption in the mask can be reduced. Another way can be optimizing the mix compositions so that the strength of concrete does not reduce significantly while using certain percentages of mask strips.

5. Figure 10a: Please write the unit of resistivity “kΩ.cm” with lowercase k (not uppercase K). Please write the unit of frequency as kHz, with lowercase k, uppercase H, and lowercase z.

Authors Reply: Thank you for your comment. This is now amended.

6. Figure 10b: Please write the unit of resistivity “kΩ.cm” with lowercase k (not uppercase K). Please write the unit of frequency as kHz, with lowercase k, uppercase H, and lowercase z.

Authors Reply: Thank you for your comment. This is now amended.

7. Figure 11a: Please write the unit of resistivity “kΩ.cm” with lowercase k (not uppercase K).  

Authors Reply: Thank you for your comment. This is now amended.

8. Figure 11b: Please write the unit of resistivity “kΩ.cm” with lowercase k (not uppercase K).  

Authors Reply: Thank you for your comment. This is now amended.

9. Figure 10 and Figure 11: Please write the relation and connection of the results in Figures 10 and 11, such as the relation of resistivity to the absorption and pore volume.

Authors Reply: Thank you for your comment. This is now amended for Figure 11. Figure 10 shows the resistivity of concrete samples for mask strips and plastic aggregates.

10. Figure 10 and Figure 11: Please also elaborate on why the trend of Figure 10a (decreasing) is “contradicting” to that of Figure 10b (increasing), similar to that of Figure 11a (decreasing) that is “contradicting” to that of Figure 10b (increasing).

 Authors Reply: The reason behind this contradicting behavior is discussed in

Section 3.5.

11. Figure 11a: Please make the y-axis to be ranged from 0 to 20%, identical to that of Figure 11b, in order to obtain quick comparison between both of them.

Authors Reply: Thank you for your comment. This is now amended.

Round 2

Reviewer 2 Report

Review of jcs-2598902.

Thank you for improving the manuscript. It can be accepted now for publication.